# Accessibility and Perceived Impact of a Group Psychosocial Intervention for Women in Ecuador: A Comparative Analysis by Migration Status

**DOI:** 10.3390/ijerph21040380

**Published:** 2024-03-22

**Authors:** Gabrielle Wimer, Maria Larrea, Josefina Streeter, Amir Hassan, Alejandra Angulo, Andrea Armijos, Annie Bonz, Wietse A. Tol, M. Claire Greene

**Affiliations:** 1Vagelos College of Physicians and Surgeons, Columbia University, New York, NY 10032, USA; ash2252@cumc.columbia.edu; 2Hunter College, New York, NY 10065, USA; maria.larrea@macaulay.cuny.edu; 3Teachers College, Columbia University, New York, NY 10027, USA; js6092@tc.columbia.edu; 4HIAS, Silver Spring, MD 62471, USA; alejandra.angulo@hias.org (A.A.); andrea.armijos@hias.org (A.A.); annie.bonz@hias.org (A.B.); 5Department of Public Health, University of Copenhagen, 1172 Copenhagen, Denmark; wietse.tol@sund.ku.dk; 6Mailman School of Public Health, Columbia University, New York, NY 10032, USA; mg4069@cumc.columbia.edu

**Keywords:** migrant, host community, MHPSS, mental health

## Abstract

There is increasing guidance promoting the provision of mental health and psychosocial support programs to both migrant and host community members in humanitarian settings. However, there is a lack of information on the respective experiences and benefits for migrant and host community members who are participating in mental health and psychosocial support programming. We evaluated a community-based psychosocial program for migrant and host community women, Entre Nosotras, which was implemented with an international non-governmental organization in Ecuador in 2021. Data on participant characteristics and psychosocial wellbeing were collected via pre/post surveys with 143 participants, and qualitative interviews were conducted with a subset (*n* = 61) of participants. All quantitative analyses were conducted in STATA, and qualitative analysis was done in NVivo. Attendance was higher for host community members. Specifically, 71.4% of host community members attended 4–5 sessions, whereas only 37.4% of migrants attended 4–5 sessions (*p* = 0.004). Qualitative analysis shows that the intervention was less accessible for migrants due to a variety of structural barriers. However, this analysis also demonstrated that both groups of women felt a greater sense of social connectedness after participating in the program and expressed gratitude for the bonds they formed with other women. Some migrant women described negative experiences with the host community because they felt as though they could not confide in host community women and speak freely in front of them. These results underscore how the migratory context influences the implementation of mental health and psychosocial support (MHPSS) programs. As humanitarian guidelines continue to emphasize the integration of host community members and displaced persons, it is critical to account for how the same intervention may impact these populations differently.

## 1. Introduction

Within Latin America and the Caribbean, political instability, violence, poverty, and climate-related emergencies have contributed to increasing regional migration in recent decades [1,2]. As of November 2023, over 7.7 million displaced persons (including refugees, asylum seekers, and migrants) had escaped the economic and political crisis in Venezuela, with over 6.5 million remaining within Latin America and the Caribbean [3]. The disproportionate burden of mental health problems among populations impacted and displaced by humanitarian emergencies is well-documented [4,5,6]. Research on the mental health consequences of forced displacement within Latin America has identified a range of risk factors for mental health problems across the migration trajectory, including exposure to trauma before and during migration as well as discrimination, lack of integration, and socioeconomic adversity following resettlement [7,8,9,10,11].

Given the global burden of psychological distress among migrants and displaced persons in humanitarian settings, there is an urgent imperative to implement scalable and sustainable mental health interventions. A growing body of evidence has substantiated the effectiveness of community-based MHPSS programs for reducing symptoms of trauma and common mental disorders among migrants [12,13,14,15], including for conflict-affected populations and forcibly displaced persons in Latin America [16,17]. However, existing barriers hinder utilization of MHPSS services among migrant populations: limited information about how to access services, social vulnerability linked to legal or migratory status, and scarce infrastructure within host country mental health systems to address the needs of displaced persons [18,19,20,21].

The recent literature also highlights the need to develop and study innovative strategies that promote the integration of MHPSS within community settings and also include populations living in host countries [22,23,24,25,26]. International guidelines and humanitarian charters have called for the provision of MHPSS services to equitably meet the mental health needs of both migrant and host communities to minimize unintended negative consequences of interventions, strengthen the capacity of host countries, and promote dignity for all vulnerable populations [22,27]. Qualitative research examining the impacts of migrant settlements on host communities indicates potential tensions arising from inequitable distribution of resources, competition for already scarce mental health services, and perceptions of physical insecurity and transmissible diseases [23,28]. However, few studies have examined the accessibility or impact of MHPSS programs that are open to migrants as well as host community members. Available evidence is primarily limited to studies comparing the prevalence of mental health problems between host and migrant communities in conflict-affected regions or to the evaluation of services originally designed for migrant populations alone [29,30,31,32,33,34,35,36,37]. However, the recent literature has affirmed elevated levels of mental health challenges and social isolation following the COVID-19 pandemic among host populations living in conflicted-affected regions [26], particularly for young women [38], underscoring the importance of improving the relevance and effectiveness of MHPSS programming for both host and migrant communities.

Despite a need to strengthen the integration of MHPSS services within community settings, a knowledge gap persists in the accessibility and effectiveness of strategies aimed at improving mental health outcomes among both migrant and host populations. To our knowledge, there is no existing study that employs a mixed methods design to compare the impact of a mental health and psychosocial intervention on migrant compared to host communities. This study explored the experiences of both migrant and host community women in Ecuador within the MHPSS program Entre Nosotras (‘among/between us’), an MHPSS program designed in coordination with an international non-governmental organization (INGO) that oversaw implementation of the program.

The intervention, Entre Nosotras, was a five-session group intervention co-designed with migrant and host community members from the study communities in Panama and Ecuador to improve mental health, sense of safety, community connectedness, and social support. The intervention focused on improving wellbeing and drew from psychoeducation, stress management, and individual and community problem solving techniques. The strategies involved in creating and implementing the intervention are detailed elsewhere [24,25]. The specific aims of this paper are to examine whether migratory status influences the accessibility or impact of MHPSS services and to assess implementation for these two groups across acceptability, feasibility, relevance, barriers and facilitators, and sustainability.

## 2. Materials and Methods

This paper focuses on results gathered in Ecuador because only migrants were included in study sites in Panama. The study sites in Ecuador spanned two locations, the urban area of Guayaquil, which is a destination for many migrants, and the rural area of Tulcán, which is a transit area for many migrants. Throughout the implementation of the program, ‘migrants’ refers to all the following: forcibly displaced persons (including refugees and asylum seekers) and people who did not identify as being forcibly displaced but who had migrated to the study community.

### 2.1. Participants and Procedures

In total, 143 women residing in Guayaquil or Tulcán participated in the trial. Participants were recruited by referral from HIAS staff, community workers, and community leaders and through community outreach by research assistants. Research assistants contacted individuals by phone or in person to provide information about the study. Interested individuals were invited to complete a screening after providing verbal consent. Women older than 18 were eligible if their distress levels were classified as moderate or below using the Kessler-6 assessment.

In Guayaquil, five of the seven groups were migrants only, and two of the groups were comprised of migrants and host community members. In Tulcán, three of the seven groups were migrants only, and four were mixed.

The primary quantitative outcome was the Personal Wellbeing Index, which includes nine dimensions of psychosocial wellbeing: life satisfaction, standard of living, health, life achievements, personal relationships, personal safety, community connectedness, future security, and spirituality/religiosity [39]. The PWI was administered before the program started as well as one week and five weeks after the program ended. Analyses were not restricted to one observation per person. Two observations were missing PWI scores, and these were excluded from the analysis.

For the qualitative portion of the analysis, up to 10 participants per study community were invited to participate in interviews after the program ended. In order to gather a range of perspectives, these women were selected using maximum variation sampling across the following categories: (1) high vs. low levels of distress at baseline; (2) high vs. low intervention attendance; (3) migrant vs. host community member; and (4) study community. The semi-structured interviews were conducted by a member of the research team in Spanish either in person or by phone and lasted approximately 45 min.

All procedures were approved by the Institutional Review Boards at Columbia University Irving Medical Center (United States), Universidad de Santander (Panamá), and Universidad San Francisco de Quito (Ecuador). The trial protocol was published and registered online (NCT05130944).

### 2.2. Analysis

#### 2.2.1. Qualitative Data

The qualitative interviews were coded by three researchers who were fluent in Spanish. The coders reviewed 12 transcripts and used a thematic analysis approach to develop a codebook that consisted of twenty-six themes and 85 subcodes. After the codebook was developed, the three researchers each coded a subset of the interviews to evaluate intercoder reliability. The researchers then pilot coded transcripts until they achieved 98.4% agreement, which required coding 10% of all transcripts. Afterwards, the researchers split up the remaining transcripts and applied study codes. After all transcripts were coded, quotes were stratified by migrants vs. community status, and memos were generated for each code based on statements made by migrant women and statements made by host community women.

#### 2.2.2. Quantitative Data

To evaluate differences in attendance between migrant and host community members, we conducted Chi-squared analyses comparing the proportion of migrant and host community women who attended zero sessions, one to three sessions, or four to five sessions. To examine the differences in trajectories of psychosocial wellbeing from the pre- to post-intervention assessments, we constructed mixed effects models that included fixed effects for community (migrant vs. host) and time as well as an interaction term. All models included a random intercept for community and participant to account for clustering.

## 3. Results

There were no significant differences in wellbeing between migrants and host community members as measured by pre- and post-intervention scores on the Psychosocial Wellbeing Index (PWI) (Table 1). This includes the subcategories of life satisfaction, standard of living, health, life achievements, personal relationships, personal safety, community connectedness, future security, and spirituality/religion, where none of the interaction coefficients were less than *p* < 0.05.

Amongst migrants, there was an increase in life satisfaction (B = 0.07; 95% CI 0.02–0.12), standard of living (B = 0.07; 95% CI 0.01–0.12), and community connectedness (B = 0.05; 95% CI 0.01–0.10). In all other subcategories, there was no significant change over time. Amongst host community women, there was no significant change observed for any psychosocial wellbeing domain assessed within this study (Table 1).

There was a significant difference in attendance by migration status, with host community status being significantly associated with attending more sessions (*p* < 0.001) (Table 1). Qualitative data analysis elucidates many of the reasons why this difference existed and highlights key differences in how host community and migrant women perceived and experienced the program, which are not evident in the quantitative data. The breakdown of interview participants can be found in Table 2. 

### 3.1. Expectations and Motivations

Prior to the start of the program, there were many women in both the migrant and host community groups who lacked a clear understanding of what Entre Nosotras entailed, which led to feelings of nervousness and curiosity. Many women also believed that they would receive financial support to participate, though this misconception was more prevalent in the migrant group. As one host community woman in Tulcán noted, “many women, women from here have said they want to join, but what happened is that they received the wrong information, they thought, and in reality we did get a bit of help for the house, Thank God, but the women wanted to join because they were told we were going to be helped financially, help us economically, that on such date they were going to give us money, so there was some controversy”.

Despite these expectations, women in both populations continued to attend the program and noted that they were motivated to continue participating because the other women provided support. They found the program to be a space where new friendships could develop; they could share laughter and de-stress, ultimately fostering a sense of camaraderie and affirmation. 

### 3.2. Perceived Impact

Social support emerged as a highly valued outcome of the program for both communities. Both groups found solace in the opportunity to share, connect, and socialize with others, creating an environment where participants could unburden themselves, relax, and enjoy companionship. Both groups expressed satisfaction with the program’s facilitators and appreciated that the women leading the program were members of their shared community. 

However, a distinct emphasis on the communal aspect of the program was present within the migrant group. These women underscored the importance of feeling like family with fellow participants, highlighting the program as a catalyst for forming meaningful friendships. A migrant woman in Guayaquil described it this way: “the companionship, being able to speak and talk about your things, that for me was the most satisfying thing”. Migrant women also highlighted that the program helped them acquire new knowledge and learn about resources available to them in the community, such as police stations or hospitals where they could seek help if needed. Migrant women reported that developing safety plans and maps during the program increased their sense of safety and perceived social support in their communities. One migrant woman in Guayaquil noted that “what I learned from the program was how to find safe sites, unsafe sites, for example, if I have a question, a problem, I can communicate with the other women and I can also help with these issues”.

Migrants also emphasized that participating in Entre Nosotras taught them to manage their emotions when coping with adversity, problem-solving skills, how to assert their rights, and motivated them to get ahead in life. A migrant woman in Guayaquil said, “Actually, everything [in the program] touched me, but one of them was that even though I’m a migrant, I have to know my rights and that no matter where I am as a woman I have rights, and as an emigrant too. This was something that well, I’m very fearful …. But now I know that yes, or rather, that we all, regardless of where we are, we do have rights and we’re going to make them count. This is something that’s imprinted in my brain”.

Host community members on the other hand highlighted the acquisition of new perspectives as a source of empowerment, which helped them understand that they were not alone in their struggles and that they could overcome them. A host community woman in Tulcán described it this way, “I would say that sometimes one thinks one is going through difficult times, but with friends that give us motivation, I mean we all go through difficult times, and it’s up to us to face them head on and have hope that new times will come. This impacted me because it taught me to not despair when times are difficult, but to have hope that something will change what’s bad”.

Both groups experienced a sense of personal improvement after participating in Entre Nosotras. Host community and migrant women described how incorporating various intervention components into their daily lives, such as problem-solving, stress-management, and breathing techniques, contributed to enhanced wellbeing and socialization. A migrant woman in Guayaquil noted that “breathing deeply, and to breathe, to take in air, these types of exercises [are] really good and I have practiced them and they’ve worked really well”. Both communities also reported that they appreciated that the program provided a break from their daily routine.

Additionally, migrant and host community women appreciated that Entre Nosotras discussed intimate partner and gender-based violence. They noted that it was relevant to their experiences as women, even if they had not personally encountered such situations. Migrant women specifically expressed satisfaction with learning more about these issues because it increased awareness about violence against women and it equipped them to assist someone in need. A migrant woman in Guayaquil noted, “And another [thing] that helped me a lot was that I have a family member someone that I’m close to who is going through issues with violence, and one woman helped me, she gave me a phone number and other things and that was really helpful”. In contrast, there were a few host community women who noted that the program content focused too much on gender-based violence and failed to address the women’s financial struggles. A host community woman in Tulcán, in response to a question about the relevance of Entre Nosotras to her and her life replied, “not really related to what one needs, what we would need, is like I said, help with work, because that’s what’s missing for us, income, or having an income, and, I don’t know, some work to have some income because we here, don’t have work”.

### 3.3. Relevance of Interventions Including Both Migrant and Host Community Women

There was also evidence of differences in migrant and host community women’s perceptions on the few occasions where participants mentioned not feeling comfortable sharing their personal experiences or struggles with members of the other group. Some migrant women described animosity or xenophobic behavior and expressed uncertainty about participating in the sessions alongside host community women because they feared a lack of confidentiality from these women. A migrant woman in Tulcán stated, “the community here of [name of community] sometimes the people reject Venezuelans”. Similarly, two migrant participants acknowledged host community rejection and hostility towards them, one of them stating she would not attend the program if she knew host community women would be present. A majority of these women were in Tulcán, which had a larger mix of migrant and host community members. 

Nevertheless, there were instances of both migrant and host community participants acknowledging the value of sharing experiences with one another. For instance, a migrant woman whose intervention group was composed only of migrant participants stated that she would have liked to meet host community women, which could have facilitated her integration process. Another migrant participant also highlighted the program’s relevance to her experience as a migrant, emphasizing how it facilitated connections with other women who shared similar struggles despite differences in their lived experiences and migration status. Likewise, a participant from the host community in Tulcán shared that the program allowed her to meet new women and realize that “we are simply women. It doesn’t matter where we come from, what nationality we are, what religion we are, we are simply all women”.

### 3.4. Barriers and Facilitators to Accessibility and Attendance

Participants from both communities shared that lack of motivation to participate or lack of interest in the program could have been a barrier to attendance, especially given some women’s incorrect perception of the program prior to their involvement, including the belief that they would be receiving economic incentives to attend. 

External factors were a significant barrier for both migrant and host communities. Scheduling conflicts were highlighted as one issue. For example, sessions coincided with women’s working hours or occurred after work when they were too tired to attend (a concern expressed more frequently by migrant participants). Similarly, whereas the issue of childcare emerged in both communities, it was particularly prominent among migrant participants, who expressed challenges attending sessions due to their responsibilities as mothers, child illness, or their inability to bring their children along to the sessions. Similarly, women from both communities emphasized the issue of household and family responsibilities, which primarily fell on them as a mother. Participants shared that having to take care of their family and maintain their household—mainly domestic duties and childcare—could hinder attendance. Host community women from Tulcán also brought attention to their unique situation as members of a rural community who needed to take care of livestock daily.

Participants from both communities expressed that women may be unable to attend the program if they must tend to their husband’s needs. Moreover, they noted the probability that some women, who may have desired to participate in the intervention, faced limitations imposed by their husbands who did not allow them to attend the sessions. Participants within both groups underscored a common “fear of the husband” and shared a past or second-hand experiences of women struggling with intimate partner violence (IPV), noting that it is likely to hinder attendance. While discussed by women in both communities, mentions of IPV were more frequent among migrant participants. A migrant woman in Tulcán noted, “They have to cook for him, attend to him. I imagine that the husband is working and arrives home tired, she has to have the food ready”. A migrant woman in Guayaquil noted that for many woman, a barrier to participation is “the fear of the husbands, there are husbands that don’t like it if their wives leave the house because they think that they will be unfaithful, so they have this fear, that no I won’t go out because of this situation, because if he finds out he’ll hit me or he can do who knows what else.“ Even when it did not personally affect them, it was noted that IPV could be a barrier. A migrant woman in Tulcán stated, “It could be that, yes, there are partners that don’t like it, there are people who have partners that control them and who don’t like that they go out, but I don’t know. For me, that’s not an issue, thank God”.

Migrant women were more likely than host community women to report barriers to attendance due to a lack of resources. For example, some migrant participants highlighted the difficulties in attending virtual sessions due to their poor internet connection or bad phone reception. On the other hand, participants struggled to attend the in-person sessions if the program was located far from their homes. Migrant women often lacked the financial means to pay for public transportation, so they commuted by foot. Thus, their attendance was largely reliant on the weather or their health. Similarly, participants from both groups explained that the availability of financial resources could impact a woman’s decision to participate in the intervention, as she was likely to prioritize finding employment and attending work over attending the program. 

Health also posed a distinct challenge to attendance, particularly for migrant participants. Numerous migrant women shared anecdotes in which they were unable to join the sessions due to health complications, at times due to sickness but primarily due to pregnancy symptoms. A migrant woman in Tulcán stated, “I came to my sessions, but others didn’t come because they were pregnant, and they had a lot of symptoms, a lot of vomiting, and they couldn’t go out”. Health problems were not reported by host community participants as a barrier to attendance.

Finally, participants within both communities described factors and elements of the intervention that they believed facilitated or could facilitate its implementation. Migrant women emphasized receiving help with childcare and financial assistance as intervention components that have helped or could help with attendance. Women who were able to bring their children to the sessions expressed relief at knowing their children were nearby during the program. One migrant woman in Tulcán even noted that her daughter helped motivate her to go, stating, “Yes, I brought her, I didn’t miss a single session, same as her, and since they gave cookies she would go running. She would tell me ‘Mama lets go,’ and I would leave running”. Similarly, women who received support through mobile/Wi-Fi services, transportation fares, and the provision of snacks acknowledged the significance of this assistance in helping them attend the sessions.

### 3.5. Sustainability

Women from both communities also took it upon themselves to spread the benefits of the program. They recounted instances where they were able to share their newfound knowledge with family, friends, and neighbors and expressed a desire to extend invitations to more individuals. A host community woman in Tulcán noted that “here the classes that you gave us were excellent to share with our neighbors”. Host and migrant women also expressed their commitment to ongoing participation, actively inviting and motivating others to join.

However, sustainability was not just defined as expanding the program by recruiting new participants, but also as reinforcing the existing bonds between participants. Most migrant and host community women described maintaining connections through WhatsApp, utilizing it for communication and occasionally arranging in-person meetings. However, technical barriers, such as sharing phones with other family members and limited internet access, have posed challenges for some. Migrant women suggested having a leader to help foster conversation and the organizing of events, including a woman in Guayaquil who said, “It’s necessary this, and I think that, making someone a facilitator, I think that we see ourselves through her and the leader too needs to motivate and connect us”.

Despite their shared desire for the program’s continuation, the focus areas for both communities differ. Migrant women emphasized material support, including food, transportation, and childcare during sessions, with the goal of aiding women in need. They also advocated for program expansion and increased recruitment. In contrast, host community women prioritized ensuring the training sessions occur in a safe location and that the program evolve to teach additional skills that could lead to income generation, such as crafts and baking.

## 4. Discussion

This study highlights the importance of mixed methods research when evaluating mental health and psychosocial support interventions in humanitarian settings. While quantitative data showed no significant differences in the impact of the program on migrant and host community women, qualitative data revealed many differences in the perceived benefits and knowledge gained from the program. The qualitative data elucidate why host community women were significantly likely to have higher attendance and shone a light on the barriers to attendance that disproportionately affected migrant women.

Migrant women most likely faced more barriers to attendance due to their greater social and economic precarity and their lack of social ties [40]. Being in a new place probably meant that migrant women were less likely to have friends and family around who could watch their children when they attended program sessions. Additionally, their migrant status increased the likelihood that if they were able to obtain work, they would have a more unpredictable schedule or would have to work longer hours [41,42].

The benefits that host community and migrant women derived from the program also varied based on how integrated they were in the community. Migrant women noted that they benefited from activities such as the mapping exercises, which is likely because they were new to the community and needed to know where important resources were. This is in line with prior research amongst IDP populations in Myanmar, where Lee et al. found high satisfaction and uptake of MHPSS referral services during the uncertainty brought about by COVID-19 [43]. Host community women, who were likely already familiar with community resources, did not comment on this aspect of the program. However, both groups appreciated the social connectedness and support they received from the program, underscoring that there are common stressors and challenges for women in Ecuador, regardless of their migration status. The fact that women also referred to the program as an opportunity to relax suggests that a key benefit of the program may be giving women a break from the myriad stressors or “daily hassles” and demands on their time that can negatively impact wellbeing [44].

Many women were disappointed, however, that Entre Nosotras did not address their need for jobs and economic security. Women from both groups emphasized that future iterations of the program should incorporate skills training, such as crafts and baking, which could allow women to develop their own source of income. The fact that these women view a MHPSS intervention as a good place to incorporate programming aimed at improving economic security points to the importance of aid groups developing structural interventions that address underlying determinants of poor mental health and wellbeing. MHPSS programs alone will not improve mental health; programming needs to be multifaceted and address the myriad issues contributing to poor mental health and wellbeing. Work by Noubani in Beirut and Beqaa, Lebanon has shown that both patients and healthcare providers view economic instability as an important cause of mental health challenges for migrant and host community populations [45,46], and these researchers propose improving job opportunities and salaries as a way of improving psychological wellbeing. The Lancet Commission on Global Mental Health and Sustainable Development goes further and states that protecting mental health and wellbeing will require advancement on all the SDGs, including SDG1, ending poverty in all its forms everywhere [47].

Structural determinants of wellbeing for women in the program also included the migration-related vulnerabilities that contribute to intimate partner violence (IPV) and subsequently hindered their ability to fully participate in the program. Women noted that they or other participants could not attend because they had caretaking obligations in their family or feared IPV as a retaliatory response from their husbands. This finding predominantly among migrant women in our program can be contextualized by the influence of economic and social precarity—a lack of legal documentation, violence along the migratory continuum, social isolation, and financial stress—on exposure to IPV among displaced women in Ecuador, as previously documented in the literature [48]. The potential role of gendered attitudes (e.g., machismo) in IPV also underscores the need to address “socio-structural determinants of mental distress” and the multifactorial, long-term marginalization of refugee women in order to improve wellbeing [49].

Despite the overwhelmingly positive experience that both migrant and host community women report while participating in Entre Nosotras, our findings do align with prior work that found that tension can exist between migrants and host communities during the implementation of mental health programs [23,28]. While prior work has documented tensions over competition for resources, during Entre Nosotras, some migrant women described negative experiences with host community women. They described feeling rejected by the host community and as though they could not confide in host community women and speak freely in front of them. Migrants are often viewed as “the other,” further distancing them from the host community and making it challenging to integrate [50], which contributes to poor wellbeing. It must be noted that these experiences depended on the composition of the groups. Migrant women in mixed groups experienced anti-Venezuelan bias, whereas women in all-migrant groups were able to bond over their shared experiences as migrants.

These migrant women’s experiences underscore the need for trust and confidentiality in group MHPSS interventions, a goal that can be more difficult to achieve when program participants come from different backgrounds and communities. This finding should not dissuade humanitarian and development groups from making interventions accessible to both migrants and host communities. When migrants have an increased sense of community and feel better integrated into their host community, it confers protective effects and mitigates the experience of discrimination and its associated consequences [51]. However, accomplishing this community cohesion does require careful planning on the organization’s part as well as preparation of participants. If underlying tensions are high, and xenophobic attitudes are prevalent, there is no guarantee that simply putting host community and migrant women together will lead to more understanding. Existing research demonstrates the importance of social cohesion and sense of belonging to migrants’ mental health and wellbeing [52,53], but further research is needed to elucidate best practices in developing inclusive MHPSS programs for migrants and host community members.

Given that many women who participated in Entre Nosotras viewed the attempt to create an inclusive and community-based intervention as successful, and all women wanted the program to expand, it is vital to reflect on the reasons for their enthusiastic support as they can suggest a way forward for other groups hoping to implement similar MHPSS interventions. There were women in mixed groups who found the experience of meeting women from different backgrounds to be a benefit of the program. The fact that migrant and host community women were involved in the development of Entre Nosotras from the beginning, and that facilitators were from the community, most likely contributed to this sense of collective ownership of the program. Entre Nosotras was also adaptable to the women’s needs, and facilitators were trained to be flexible so that their activities could best support the specific group of women they were working with. These aspects of the program allowed for strong bonds to form between the women, many of whom have continued to meet and correspond via WhatsApp. This multimodal communication points to the benefits of a hybrid program, especially when it comes to sustainability. However, this was not initially a choice made by the implementors of the program; the COVID-19 situation at the time in Ecuador necessitated it.

It is also important to note the other ways in which COVID19 could have impacted the program. Given that people were feeling particularly isolated at this time, it could have led to an increase in the sense of community that would not be present if people were able to engage in their daily routines [54,55]. Other limitations to our analysis are that participants opted in, so host community women were more likely to be comfortable working and meeting with migrant women, and vice versa. In other programs where participants do not self-select in this way, it is probable that tensions between migrants and host community women would be even higher. There was also likely an element of social desirability bias in interviews because the research team was the same team who trained facilitators and helped implement Entre Nosotras [56]. This could have led women to overstate the benefits of the program and their satisfaction with Entre Nosotras.

Lastly, this analysis primarily focuses on the context in Ecuador between Ecuadorians and Venezuelans (though there were a small number of migrants from other countries). Thus, our findings are not necessarily generalizable to other contexts or migrant and host populations. However, our focus on Latin America does fill a gap in the existing literature examining MHPSS interventions for both migrants and host community members that has primarily focused on Africa, Asia, and the Middle East [23,26,30,32,43,45,46,57,58,59,60,61,62]. Additionally, these other studies did not primarily focus on women. Future qualitative research on mental health of women should probe the links among legality, financial precarity, and gender-based violence. As our analysis shows, women face unique barriers and challenges; therefore, examining and understanding their experiences accessing and participating in MHPSS interventions are critical.

## 5. Conclusions

To the best of our knowledge, this is the first study looking at an MHPSS intervention for migrant and host community women that uses a mixed methods analytic approach, and it is the only one to do so in Latin America. Our analysis has shown the importance of incorporating qualitative analysis because it elucidates how barriers can vary for migrants and host community members and that the perceived benefits and relevance of an intervention can vary by migration status as well.

As international guidelines and humanitarian charters continue to emphasize the importance of making MHPSS interventions and programs available to host communities as well as migrants, further research is needed to minimize any unintended negative consequences of interventions. Further research should specifically continue to explore the dynamics between migrant and host communities in Latin America where political instability and poverty are key drivers of migration as opposed to conflict. Moving forward, MHPSS interventions in other settings could adapt a similar evaluation framework to understand how local context and country of origin impact utilization, acceptability, feasibility, and impact for migrant and host community groups.

## Figures and Tables

**Table 1 ijerph-21-00380-t001:** Trajectories in wellbeing and attendance patterns by migratory status.

	Migrant (REF)	Host Community Member	
	Mean Change (95% CI)	Mean Change (95% CI)	Interaction z (*p*)
Overall wellbeing	0.24 (0.04, 0.45)	−0.01 (−0.40, 0.37)	−1.14 (0.253)
Life satisfaction	0.07 (0.02, 0.12)	0.02 (−0.07, 0.11)	−1.01 (0.315)
Standard of living	0.07 (0.01, 0.12)	0.08 (−0.02, 0.18)	0.28 (0.777)
Health	0.02 (−0.03, 0.07)	0.00 (−0.10, 0.09)	−0.37 (0.710)
Life achievements	0.04 (−0.01, 0.08)	0.00 (−0.09, 0.08)	−0.89 (0.371)
Personal relationships	0.03 (−0.02, 0.07)	0.00 (−0.09, 0.09)	−0.51 (0.609)
Personal safety	0.03 (−0.02, 0.07)	−0.05 (−0.13, 0.03)	−1.62 (0.105)
Community connectedness	0.05 (0.01, 0.10)	0.00 (−0.09, 0.09)	−1.00 (0.316)
Future security	0.03 (−0.01, 0.06)	−0.01 (−0.09, 0.06)	−0.87 (0.387)
Spirituality/religion	−0.01 (−0.05, 0.03)	−0.02 (−0.08, 0.05)	−0.15 (0.877)
Number of sessions attended	N (%)	N (%)	Chi2 = 10.90 (*p* = 0.004)
0	35 (30.43)	5 (17.86)	
1–3	37 (32.17)	3 (10.71)	
4–5	43 (37.39)	20 (71.43)	

**Table 2 ijerph-21-00380-t002:** Number of participants included in qualitative analysis.

	# of Women from Host Community	# of Migrant Women
Guayaquil	4	27
Tulcán	6	24
Total	10	51

## Data Availability

The data that support the findings of this study are available from the corresponding author, M.C.G., upon reasonable request.

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
