# Peer review of "Accessibility and Perceived Impact of a Group Psychosocial Intervention for Women in Ecuador: A Comparative Analysis by Migration Status"

_ijerph, 2024, doi:10.3390/ijerph21040380_

Round 1
Reviewer 1 Report
Comments and Suggestions for Authors
This paper provides a good addition to the literature on migrant well-being interventions by looking at the perceptions of both migrant and host women participating in well-being programs. It is well-written and clear, the methodology is appropriate, the results described in sufficient detail and the discussion raises issues of relevance to developers and researchers working with migrant communities, particularly in South America. While the quantitative results reveal few differences between the perceptions of the two groups, the interview data is used to throw light on a number of important areas for consideration in the design of future interventions. I recommend publication after some clarification and additional details as outlined below:
· The composition of the groups needs to be made clear from the beginning. It emerges at various points in the paper that some groups were mixed, while others were made up solely of migrant or host women. There needs also to be acknowledgement that this may have had different effects on the women’s experience of the intervention and some discussion of the pros and cons of mixed or single focus groups. How women were recruited to the program would also be useful information.
· There is some confusion around the numbers of women participating in the intervention. The Abstract has 143 participants (as does p.3). The N in Table 1 for ‘Number of sessions attended’ is 439 in total. If the difference is due to rates of completion of the PWI, then this needs to be made clear, with information on missing data and how this was handled in the analyses. The abstract states that 67 interviews were conducted, but the interview number in table 2 is 61. Again, this difference needs to be explained.
Author Response
The composition of the groups needs to be made clear from the beginning. It emerges at various points in the paper that some groups were mixed, while others were made up solely of migrant or host women.
We have incorporated the demographic breakdown of the groups into the methods section, lines 112-114.
There needs also to be acknowledgement that this may have had different effects on the women’s experience of the intervention and some discussion of the pros and cons of mixed or single focus groups.
Thank you for this important point, we have incorporated this information into the discussion section, lines 433-435 and 456-460.
How women were recruited to the program would also be useful information.
We have included this information in the methods section, lines 105-109.
There is some confusion around the numbers of women participating in the intervention. The Abstract has 143 participants (as does p.3). The N in Table 1 for ‘Number of sessions attended’ is 439 in total. If the difference is due to rates of completion of the PWI, then this needs to be made clear, with information on missing data and how this was handled in the analyses.
We did not restrict that part of the analysis to one observation per person. We have also updated the manuscript to reflect that 2 observations were missing PWI scores and were excluded from the analysis. Please see lines 118-121.
The abstract states that 67 interviews were conducted, but the interview number in table 2 is 61. Again, this difference needs to be explained.
Thank you for pointing out this mistake. The abstract has been updated to reflect the correct number of 61 interviews.
Reviewer 2 Report
Comments and Suggestions for Authors
The article presents the existing literature on MHPSS in a concise and efficient manner, pinpointing a gap in the literature to be filled by the present study. Methodology section is also effective in presenting the data collection and analysis methods used for the study.
The findings are not particularly revealing, but valuable nonetheless. Numerous sociological and anthropological studies have shown that psychological wellbeing is strongly correlated to financial stability. For distressed population groups, such as the migrants this study examined, it is no surprise to learn that what they most wanted was not the techniques to reduce the stress, but the measures to address the causes of their stress itself (i.e., legal and financial precarity). It is telling that many migrant women in the study highlighted the social connections they built with other migrant women is the biggest benefit, rather than the content of Entre Nuestra's programming.
While I appreciated the careful interpretation of the qualitative data (i.e., interview narratives), I wished the authors could have interrogated the link between legality, financial precarity, and gender-based violence more critically, instead of treating them as independent factors that are impacting the women's mental wellbeing. Cultural expectations about gender roles and gendered attitudes (e.g., machismo) may indeed be an important factor, but one must examine IPV, for instance, in relation to the overwhelming amount of financial stress and anxiety, partly caused by their precarious legal status, they, as family units, are experiencing. Without the rigorous intersectional theorization, the study like this can easily fall into cultural determinism, and reaffirmation of an existing harmful stereotype of Latin American men as somehow inherently violent and sexist, due to their "culture."
Author Response
Thank you very much for this important critique. We have reformulated our paragraph on IPV to better reflect the intersectionality of these issues (lines 419-430) and we have included a call for more research into this in our conclusion (lines 491-493)